# Machine Learning Model-Based Simple Clinical Information to Predict Decreased Left Atrial Appendage Flow Velocity

**DOI:** 10.3390/jpm12030437

**Published:** 2022-03-10

**Authors:** Chao Li, Guanhua Dou, Yipu Ding, Ran Xin, Jing Wang, Jun Guo, Yundai Chen, Junjie Yang

**Affiliations:** 1Chinese PLA Medical School, Haidian District, Beijing 100039, China; imaginemedicine@163.com; 2Chinese PLA General Hospital, Haidian District, Beijing 100039, China; guanhuadou@163.com (G.D.); Wangjing121@126.com (J.W.); 3School of Medicine, Nankai University, Tianjin 300071, China; 2120201296@mail.nankai.edu.cn (Y.D.); nkuxinran@163.com (R.X.); 4Department of Cardiology, The Sixth Medical Center of PLA General Hospital, Haidian District, Beijing 100039, China; guojun301@163.com (J.G.); fearlessyang@126.com (J.Y.)

**Keywords:** atrial fibrillation, left atrial appendage, machine learning, flow velocity

## Abstract

Background: Transesophageal echocardiography (TEE) is the first technique of choice for evaluating the left atrial appendage flow velocity (LAAV) in clinical practice, which may cause some complications. Therefore, clinicians require a simple applicable method to screen patients with decreased LAAV. Therefore, we investigated the feasibility and accuracy of a machine learning (ML) model to predict LAAV. Method: The analysis included patients with atrial fibrillation who visited the general hospital of PLA and underwent transesophageal echocardiography (TEE) between January 2017 and December 2020. Three machine learning algorithms were used to predict LAAV. The area under the receiver operating characteristic curve (AUC) was measured to evaluate diagnostic accuracy. Results: Of the 1039 subjects, 125 patients (12%) were determined as having decreased LAAV (LAAV < 25 cm/s). Patients with decreased LAAV were fatter and showed a higher prevalence of persistent AF, heart failure, hypertension, diabetes and stroke, and the decreased LAAV group had a larger left atrium diameter and a higher serum level of NT-pro BNP than the control group (*p* < 0.05). Three machine-learning models (SVM model, RF model, and KNN model) were developed to predict LAAV. In the test data, the RF model performs best (R = 0.608, AUC = 0.89) among the three models. A fivefold cross-validation scheme further verified the predictive ability of the RF model. In the RF model, NT-proBNP was the factor with the strongest impact. Conclusions: A machine learning model (Random Forest model)-based simple clinical information showed good performance in predicting LAAV. The tool for the screening of decreased LAAV patients may be very helpful in the risk classification of patients with a high risk of LAA thrombosis.

## 1. Background 

Atrial fibrillation is the most common arrhythmia in clinical practice and is associated with stroke [1]. As shown in previous reports, the left atrial appendage (LAA) may harbor up to 90% of thrombi occurring in patients with AF [2].

The left atrial appendage flow velocity (LAAV) can reflect left atrial appendage function, which has many clinical implications. First, a low LAAV indicates a high risk of LAA spontaneous echo contrast (SEC) and LAA thrombus [3], and SEC is an independent risk factor for subsequent thromboembolic events [4]. Second, LAAV was proved to be an independent predictor of cardioversion success [5]. The measurement of LAAV could provide useful information for the prediction of cardioversion outcomes in AF patients. Third, previous studies have proved that a low LAAV was associated with AF recurrence after the initial catheter ablation of persistent AF [5]. Moreover, low flow velocity in LAA is a predisposing factor in AF patients [6]. 

Now, transesophageal echocardiography (TEE) is the first technique of choice to evaluate the LAAV [7]. Although TEE is relatively safe and noninvasive, the insertion and manipulation of the ultrasound probe may cause some complications such as oropharyngeal, esophageal, or gastric trauma [8,9]. Therefore, clinicians require a simple applicable method with high sensitivity, which could screen patients with decreased LAAV. 

Machine learning (ML) has been used and has shown an acceptable performance in predicting the risk of diseases [10,11,12]. Therefore, in the current study, we used simple clinical data obtained by an ML model to predict LAAV and aim to examine the feature importance of the ML model to understand its mechanism.

## 2. Methods 

### 2.1. Participants

In the present study, we have retrospectively recruited patients who visiting the General Hospital of PLA and underwent transesophageal echocardiography in between January 2017 and December 2020. Inclusion criteria: (1) age ≥ 18 years old; (2) data documented in HIS system in hospital; (3) complete transesophageal echocardiography examination; (4) diagnosed with nonvalvular atrial fibrillation. Subjects were excluded if they did not meet the above criteria.

### 2.2. Transesophageal Echocardiography 

Left atrial appendage flow velocity was measured by TEE, and all transesophageal echocardiography procedures were performed by experienced cardiologists. Evaluation of LAA by TEE included the presentation or absence of left atrial thrombus, left atrial appendage spontaneous echo contrast, and decreased left atrial appendage emptying peak flow velocity. Based on previous research, LAAV of <25 cm/s was suggested as a useful cutoff value to discriminate patients with a high risk of systemic embolization [13,14]. 

### 2.3. Clinical Factors 

All clinical data were searched for by the HIS system (Hospital Information System). Since the model we built is a primary screening model, covariates for the machine leaning model were chosen based on prior literature review and clinical judgment, focusing on easily accessible variables that were expected to affect the LAAV. 

Predictors in our model included demographics (age and sex), previous medical history (history of atrial fibrillation, hypertension, heart failure, diabetes, stroke, and vascular diseases), anticoagulant drugs and left atrial (LA) diameter. In addition, many studies have proved the correlation between NT-proBNP and LAA flow velocity [15,16,17]. Therefore, we chose BNP as one of the predictors.

To ensure accuracy, all the data were collected twice. If there was any inconsistency, a third review was conducted. 

### 2.4. Study Design 

As shown in Figure 1, the data were randomly split into a training set (80%) and a test set (20%). The CreateDataPartition function (caret package) in the R 15.6 Available online: http://www.R-project.org (accessed on 11 February 2022). was used to segment the dataset, and a statistical test was carried out to ensure the balance of the variables in the two datasets. A training set was used for the training of the ML model, and the models were tested in the testing set by cross-validation and 5-fold validation.

### 2.5. Model Development 

In the current study, we built regression models using various ML algorithms, including the support vector machine (SVM) model, random forest (RF) model, and k-nearest neighbor (KNN) model because they are used widely in regression models [18,19,20,21]. All models were developed by R 15.6. 

### 2.6. Model Evaluation 

The optimal cut-off point of the predicted LAAV was determined by receiver operating characteristic curve (based on the principle of maximum Yordon index), using LAAV measured by TEE as a gold standard (<25 cm/s). With the cut-off values, the sensitivity, specificity, and F1 score of three models were calculated. Accuracy, sensitivity, specificity, mean squared error and area under the receiver operating characteristic curve (AUCROC), and F1 score were used to evaluate the performance of the model. Accuracy refers to the ratio of the number of correctly predicted decreased LAAV to the total number of participants. Among the metrics, AUCROC and F1 score were the main metrics to reflect the performance of the model. 

### 2.7. Fivefold Cross-Validation 

Considering that the model may suffer from overfitting, we performed a fivefold cross-validation scheme. First, the data were partitioned into 5 equal parts. The model was trained on 4 parts, leaving 1 part for testing. The process was repeated 5 times until testing was performed on all of the 5 parts.

## 3. Definition and Statistics of Variables 

Continuous variables (age, Body Mass Index (BMI), serum of NT-proBNP and left atrial diameter) are presented as medians (interquartile ranges) and compared by Mann–Whitney U tests. Categorical variables (gender, anticoagulant, history of persistent AF, heart failure, hypertension, diabetes, stroke, and vascular disease) were presented as number (proportion) and compared by Pearson’s χ^2^ test or Fisher’s exact test. A *p*-value of less than 0.05 was considered significant. 

## 4. Results 

### 4.1. Patient Characteristics 

Of the 1152 patients undergoing TEE, 1039 patients were included in the final analysis (Figure 1). Among the 1039 patients, 125 patients (12%) were diagnosed with decreased LAAV (<25 cm/s). 

As shown in Table 1, patients with low LAAV flow showed a higher prevalence of persistent AF (66.4% vs. 27.7%, *p* < 0.001), heart failure (32.8 vs. 7.3%, *p* < 0.001), hypertension (63.2% vs. 51.9%, *p* = 0.011), diabetes (31.2% vs. 21.9%, *p* = 0.015), and stroke (21.6% vs. 13.9%, *p* = 0.02). Left atrial diameter (LA diameter) was larger in patients with decreased LAAV than in those with normal LAAV (35–43 vs. 35–42, *p* < 0.001), and patients with decreased LAAV were fatter than those in the control group (BMI 26.9 vs. 25.8, *p* = 0.01). The serum level of NT-pro BNP was higher in the decreased-LAAV group than in the normal-LAAV group (Table 1). 

### 4.2. The Characteristics of the Training Set and Testing Set

As shown in Table 2, of the 1039 patients, 834 patients (80%) were included in the training set, and the remaining 259 patients (20%) were included in the test set. The ratio of decreased LAAV was not significantly different in the training set compared to the test set (11.6% vs. 13.7%, *p* = 0.424). Patients in the training set showed a similar prevalence of persistent AF (31.7% vs. 35.1%, *p* = 0.342), heart failure (9.8% vs. 12.7%, *p* = 0.231), hypertension (53.8% vs. 50.7%, *p* = 0.425), diabetes (22.9% vs. 23.4%, *p* = 0.876), stroke (14.9% vs. 14.6%, *p* = 0.933), and vascular disease (9.4% vs. 6.8%, *p* = 0.255). The LAAV was not significantly different in the training set than in the testing set (*p* = 0.727).

### 4.3. Development of Machine Learning Model 

The development of the ML models is shown in Figure 2. In the KNN model, the K value was adjusted to develop the best model, and the performance of the KNN model suggested the best accuracy with a K value of 49 (the absolute error: 13.4 cm/s). The development of the RF model and the KNN model is shown in Figure 2. In the RF model, the number of trees was 500. The support vector machine (SVM) model was also developed based on the training dataset.

### 4.4. Model Comparison for Regression and Binary Classification Problem

The ability of the three ML models to predict the LAAV is shown in Table 3 (training set) and Table 4 (testing set). The RF model performed the best of the three models. In the testing set, the root-mean-square errors (RMSEs) of the three models were 17.51 cm/s (KNN model), 16.65 cm/s (RF model) and 17.66 cm/s (SVM model), and the mean absolute error of the RF model was also the smallest among the three models (13.4 cm/s, 13.04 cm/s and 13.68 cm/s, respectively). 

The ability of the ML algorithms to discriminate between decreased LAAV and normal LAAV is shown in Table 3 and Table 4.

In the training set, the KNN model showed the poorest performance, with an AUCROC of 0.81 (0.76–0.85), while the RF model had the highest AUCROC of 0.98 (0.97–0.99) (RF model vs. KNN model, *p* < 0.001; RF model vs. SVM model, *p* < 0.001). The cut-off values were added in Table 3. With the cut-off values, the sensitivity, specificity, accuracy, and F1 score of the three models were calculated. The RF model has the best accuracy (92%) and highest F1 score among three models (KNN 0.44; RF 0.77; SVM 0.58).

In the testing set, the KNN model showed the poorest performance, with an AUCROC of 0.81 (0.73–0.89), while there was no difference found in AUCROC between the RF model and SVM model (*p* = 0.373). As shown in Table 4, although the sensitivity of the RF model (81%) was lower than the KNN model (93.5%) and SVM model (100%), the specificity of the RF model was highest (RF 86%, KNN 58% and SVM 62%) with the highest accuracy of 85% and highest F1 score of 0.62 (KNN 0.43, SVM 0.48). 

The calibration plots of the models were shown in Figure 3. A Hosmer–Lemeshow test was used to evaluate the classification efficiency of the three models, and *p*-values were shown in Figure 3. All models indicated appropriate calibration (*p* > 0.05).

As reported in previous studies, a model with AUC ranging from a range of 0.80 to 0.90 is considered excellent [22]. The RF model showed acceptable AUC in both training and testing sets (0.98 and 0.89) and performed better than other non-invasive methods using CT [14,23].

The results of the fivefold cross-validation showed that the RF model showed a better discriminative ability for decreased LAAV than the other two models (Table 5).

### 4.5. Factors Predicting Decreased LAAV in the RF Model

The overall attributions of variables in the Random Forest model are shown in Figure 4. 

The percentages of increase in mean square error (%IncMSE) and increased node purity (IncNodePurity) were used to evaluate the importance of each variable in the model. The serum level of NT-pro BNP contributed the most (%IncMSE: 34.5) to the LAAV prediction, followed by the diagnosis of persistent AF (%IncMSE: 28.0), LA size (%IncMSE: 25.6%), BMI (%IncMSE: 14.7), weight (%IncMSE: 13.3%), etc. IncNodePurity relates to the loss function, which is chosen by best splits. The loss function is MSE for regression and Gini-impurity for classification. In the RF model, the serum level of NT-pro BNP, LA size, BMI, and diagnosis of persistent AF played an important role with high IncNodePurity in all variables.

## 5. Discussion 

We developed three machine learning models (KNN model, Random Forest model and SVM model) to predict LAAV in AF patients using simple clinical risk factors. The models were trained using data from 834 patients and tested using data from 205 patients. In this retrospective analysis, the RF model demonstrated the highest accuracy (AUCROC: 0.89, MAE: 13.04 cm/s) of the three models when validating in the testing set. In the RF model, the serum level of NT-pro BNP contributed the most to the LAAV prediction, followed by LA size, diagnosis of persistent AF, BMI, weight, etc.

The left atrial appendage was proved to be a major source of emboli responsible for cardioembolic stroke in previous studies [24], and decreased LAAV has been proved to be a well-recognized risk factor for left atrial appendage thrombosis and stroke [6,24,25,26]. In clinical practice, TEE was regarded as the first technique of choice to measure the function of left atrial appendage, but TEE may cause severe discomfort in patients and, moreover, serious complications (such as esophageal damage) [27]. Freitas-Ferraz et al. conducted a study including 1249 consecutive patients undergoing TEE and found that the overall incidence of TEE-related complications was 0.9% to 6.1% [28]. Therefore, clinicians require a noninvasive method to screen patients with decreased LAAV. 

Coletta et al. [23] proved that transthoracic echocardiography (TTE) could be used to identify patients with low and high blood flow velocities, but only 84% of the patients could be measured by TTE, and this study was conducted with a small number of patients (86 patients). Yasuoka et al. [14] found another noninvasive method to predict LAAV using enhanced computed tomography. They found that the LAAV could be estimated by the HU density ratio at distal and proximal sites within the LAA. However, the study was conducted with a small number of patients (60 patients) and, in clinical practice, the HU density ratio of many patients could not always be measured because of poor-contrast filling of the left atrial appendage. In this study, using noninvasive clinical data, we developed an ML model to predict the LAAV of AF patients, which could be used to primarily screen patients with decreased LAAV in a cheap and fast way.

Machine learning has shown acceptable results in various medical fields. In ML model development, to avoid overfitting of the model, it has been suggested that the number-to-feature ratio should be at least 5 [29]. In the present study, the ratio was about 70. The data were randomly divided into two subsets, and the features in the testing set were not significantly different than the training set. In cross-validation, the RF model had 86% specificity and 81% sensitivity. We further assessed the model by fivefold cross-validation, and, consistently with the previous results of cross-validation, the RF model performed best (AUVROC 0.85).

Goldman et al. found that age, systolic blood pressure, sustained AF, ischemic heart disease, and left atrial area were associated with LAAV [30]. Demircelik et al. found that left ventricular diastolic dysfunction was associated with left atrial appendage functions [31]. Handke et al. found that left ventricular ejection fraction, LA size, paroxysmal AF, age, and sex are independent parameters influencing LAAV [32]. Although many clinical factors associated with decreased LAAV are increasingly available, risk estimation of decreased LAAV remains challenging. The indicators included in our model were easily accessible; therefore, our model may enable instantaneous risk estimation of decreased LAAV, which may facilitate rapid identification of individuals at elevated risk to guide further invasive inspection.

Although machine learning models may be useful to help us to diagnose patients, they are still a “black box”, lacking acceptable interpretability. In the RF model, we further uncovered important predictors of decreased LAAV. Based on our results, serum level of NT-proBNP and LA size was the most important variable (Figure 4), which was also confirmed as a risk factor in previous research [16,33]. The percentage of IncMSE (% increased mean square error) was equivalent to mean decrease accuracy, which could be used to measure the importance of each variable in the model. Other factors, such as diagnosis of persistent AF, weight, and BMI, were still proven to be useful for predicting LAAV. Because machine learning models can quickly process large amounts of data, we note other implications of our study, which suggest that machine learning models may contribute to extract elements of risk. 

It is worth noting the use of anticoagulants in AF patients. A previous study has shown that the underuse of OAC is common [34]. In 2008, only fewer than half were treated with anticoagulation (2.7%, warfarin; 39.7%, aspirin) [35]. In 2016, Xiong et al. found that only 44.5% of Chinese AF patients received OAC treatment in a research-based Chinese population [36]. In recent years, the use of anticoagulants has gradually increased but remains inadequate. In our results, about half of the AF patients (50.8%) were not taking anticoagulants, which indicates that standardized anticoagulant use remains inadequate.

In our model, the serum level of NT-proBNP was one of the most important variables. Several mechanisms may explain the association between NT-proBNP and decreased LAAV. BNP is secreted mainly from the left atrium, and atrial pressure overload leads to elevation in plasma BNP levels [37]. Previous studies [38,39] found that elevated BNP was associated with LAA disfunction, and a high plasma brain natriuretic polypeptide level was a marker of risk for thromboembolism in patients with nonvalvular atrial fibrillation [40]. Harada et al. also found that higher plasma BNP was associated with a lower LAA flow velocity in patients with nonvalvular AF and normal LV systolic function [17]. Therefore, in addition to diagnosing heart failure, the serum level of NT-pro BNP could, to a certain degree, reflect the function of left atrium and left atrial appendage. Therefore, it plays a great role in predicting the blood flow velocity of the left atrial appendage. 

## 6. Limitations 

This study has some limitations. First, this study was a single-center and retrospective study, and had a limitation in generalizability. Second, the model was only tested in the test set instead of a prospective cohort in the real world. Third, there may be some missing clinical features in the candidate features which may contribute to the improvement of the model. 

## 7. Conclusions 

Machine learning model (Random Forest model)-based simple clinical information showed good performance in predicting LAAV. The tool for screening decreased LAAV patients may be very helpful in the risk classification of patients with a high risk of LAA thrombosis. 

## Figures and Tables

**Figure 1 jpm-12-00437-f001:**
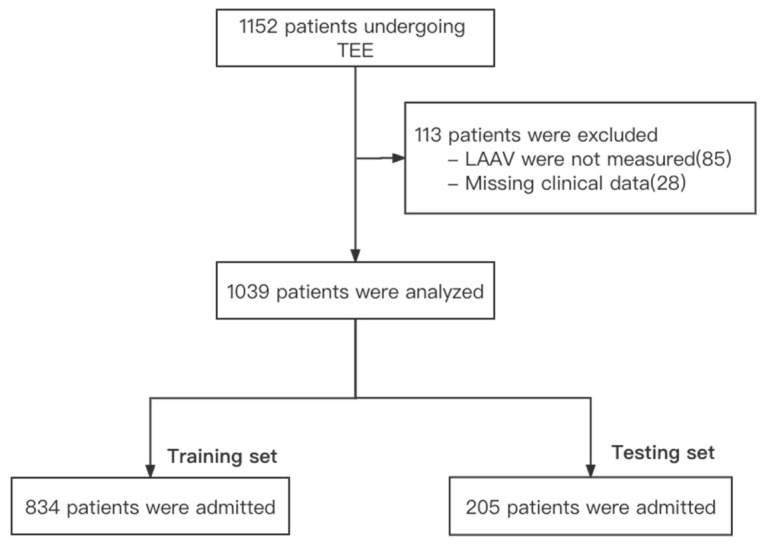
Flow diagram. TEE: transesophageal echocardiography; LAAV: left atrial appendage flow velocity.

**Figure 2 jpm-12-00437-f002:**
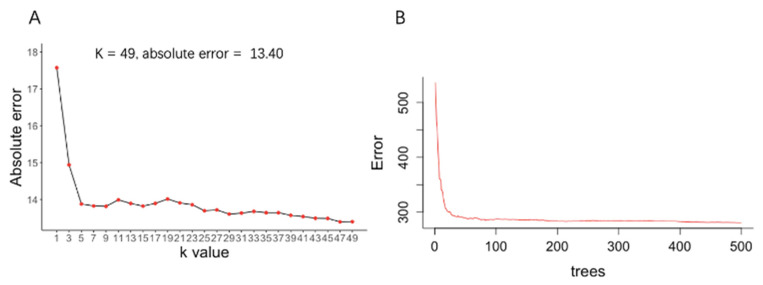
Development of KNN Model and Random Forest Model. (**A**) Development of KNN model; (**B**) Development of Random Forest model.

**Figure 3 jpm-12-00437-f003:**
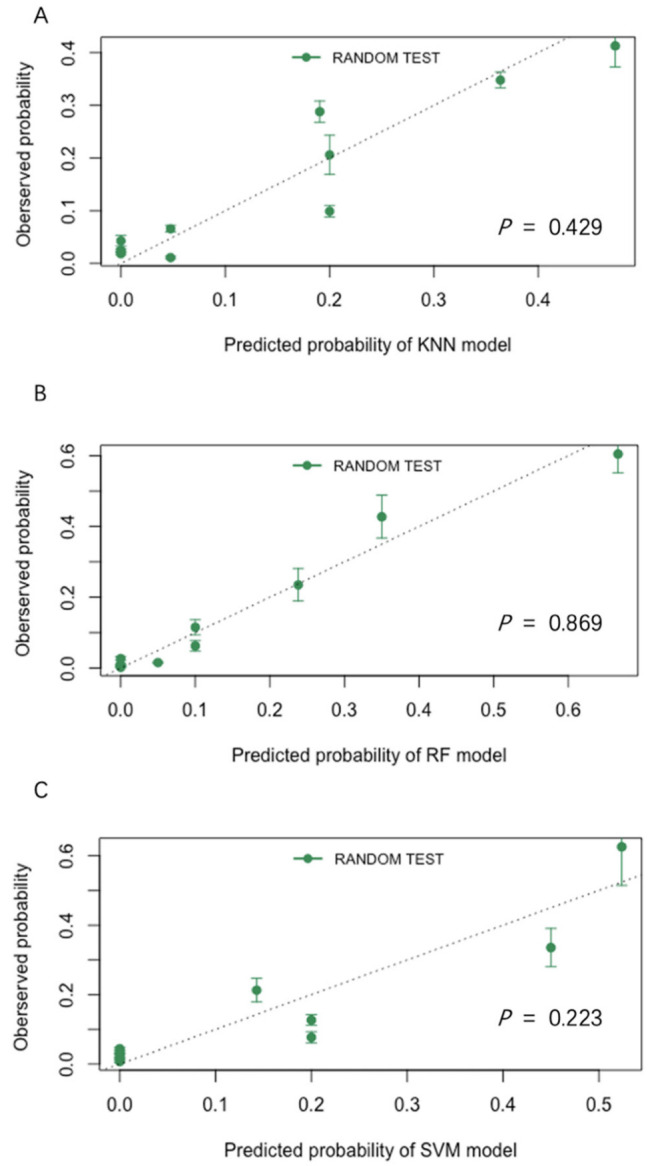
Calibration plots for the KNN(A), RF(B), and SVM(C) models. (KNN, k-Nearest Neighbor; RF, random forest; SVM, support vector machine). All models indicated appropriate calibration (*p* > 0.05). (**A**) Calibration plot for the KNN model; (**B**) Calibration plot for the RF model; (**C**) Calibration plot for the SVM model.

**Figure 4 jpm-12-00437-f004:**
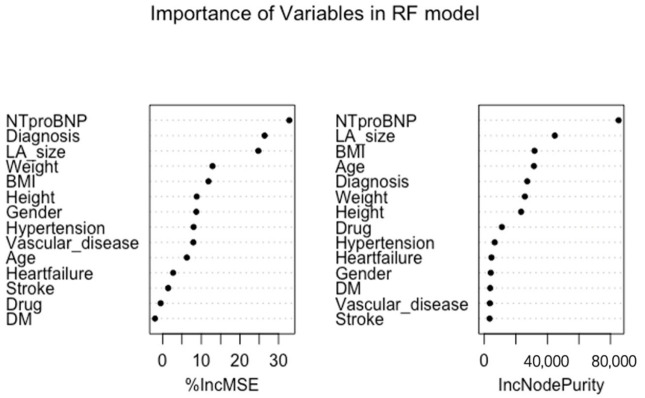
The importance of variables in Random Forest model. BMI: body mass index; DM: diabetes mellitus.

**Table 1 jpm-12-00437-t001:** Baseline characteristics of the records of the enrolled patients with decreased LAAV and normal LAAV.

	ALL *n* = 1039	Normal LAAV *n* = 914	Decreased LAAV *n* = 125	*p*-Value
Age (years)	62 (54–69)	62 (54–68)	64 (54–71.5)	0.022
Gender (male)	757 (72.9%)	663 (72.5%)	94 (75.2%)	0.530
BMI (kg/m^2^)	26 (23–28)	25.8 (23.8–28.0)	26.9 (24.4–29.3)	0.010
Persistent AF	336 (32.3%)	253 (27.7%)	83 (66.4%)	<0.001
Antithrombotic therapy				<0.001
No antithrombotic therapy	528 (50.8%)	478 (52.3%)	50 (40.0%)	
Aspirin	130 (12.5%)	113 (12.4%)	17 (13.6%)	
Clopidogrel	11 (1.1%)	8 (0.9%)	3 (2.4%)	
Dual antiplatelet	21 (2.0%)	18 (2.0%)	3 (2.4%)	
Warfarin	95 (9.1%)	76 (8.3%)	19 (15.2%)	
NOAC	254 (24.5%)	221 (24.2%)	33 (26.4%)	
Heart failure	108 (10.4%)	67 (7.3%)	41 (32.8%)	<0.001
Hypertension	553 (53.2%)	474 (51.9%)	79 (63.2%)	0.017
Diabetes	239 (23.0%)	200 (21.9%)	39 (31.2%)	0.020
Stroke	154 (14.8%)	127 (13.9%)	27 (21.6%)	0.023
Vascular disease	92 (8.9%)	83 (9.1%)	9 (7.2%)	0.487
NT-pro BNP (pg/mL)	319 (116–770)	266 (103–635)	997 (590.5–1862)	<0.001
LA diameter (mm)	39 (35–43)	38 (35–42)	43 (40–47.5)	<0.001

NOAC: New oral anticoagulants, include dabigatran and rivaroxaban; LA diameter: left atrial diameter; BMI: Body Mass Index.

**Table 2 jpm-12-00437-t002:** The characteristics of training set and testing set.

Feature	Training Set *n* = 834	Testing Set *n* = 205	*p*-Value
Age (years)	62 (54–69)	61 (54–68.5)	0.357
Gender(male)	592 (71.0%)	165 (80.5%)	0.006
BMI (kg/m^2^)	26 (23.9–28.1)	26 (24.2–28.2)	0.772
Persistent AF	264 (31.7%)	72 (35.1%)	0.342
Antithrombotic therapy			0.68
No antithrombotic therapy	426 (51.2%)	102 (49.8%)	
Aspirin	104 (12.5%)	26 (12.7%)	
Clopidogrel	10 (1.2%)	1 (0.5%)	
Dual antiplatelet	19 (2.3%)	2 (1%)	
Warfarin	72 (8.6%)	23 (11.2%)	
NOAC	203 (24.3%)	51 (24.8%)	
Heart failure	82 (9.8%)	26 (12.7%)	0.231
Hypertension	449 (53.8%)	104 (50.7%)	0.425
Diabetes	191 (22.9%)	48 (23.4%)	0.876
Stroke	124 (14.9%)	30 (14.6%)	0.933
Vascular disease	78 (9.4%)	14 (6.8%)	0.255
NT-pro BNP (pg/mL)	309 (111–725)	401 (150–885)	0.061
LA size (mm)	39 (35–43)	39 (36–42)	0.446
LAAV (cm/s)	46 (33–60.25)	46 (36–42)	0.727
Decreased LAAV	97 (11.6%)	28 (13.7%)	0.424

**Table 3 jpm-12-00437-t003:** Comparison of the predictive performance for three models in the training set.

Model	Accuracy (%)	F1 Score	AUC	Cut-off Value	R^2^	RMSE (cm/s)	MAE (cm/s)
KNN	76	0.44	0.81 (0.76–0.85)	41.3	0.26	17.29	13.80
RF	92	0.77	0.98 (0.97–0.99)	33.8	0.85	7.85	5.97
SVM	84	0.58	0.91 (0.87–0.94)	37.1	0.23	17.66	13.68

KNN, k-nearest neighbor; RF, random forest; SVM, support vector machine; AUC, area under the receiver operating characteristic curve; R^2^, R Squared; RMSE, root-mean-square error; MAE, mean absolute deviation.

**Table 4 jpm-12-00437-t004:** Comparison of the predictive performance for three models in the testing set.

Model	Accuracy (%)	F1 Score	AUC	Specificity (%)	Sensitivity (%)	R^2^	RMSE (cm/s)	MAE (cm/s)
KNN	63	0.43	0.81 (0.73–0.89)	58	93.5	0.24	17.51	13.40
RF	85	0.62	0.89 (0.83–0.95)	86	81	0.31	16.65	13.04
SVM	67	0.48	0.87 (0.82–0.93)	62	100	0.23	17.66	13.68

KNN, k-nearest neighbor; RF, random forest; SVM, support vector machine; AUC, area under the receiver operating characteristic curve; R^2^, R Squared; RMSE, root-mean-square error; MAE, mean absolute deviation.

**Table 5 jpm-12-00437-t005:** Comparison of the predictive performance for three models (fivefold cross-validation).

Model	AUCROC	R^2^	RMSE (cm/s)	MAE (cm/s)
KNN	0.806	0.24	17.43	13.40
RF	0.854	0.735	10.28	7.44
SVM	0.84	0.39	15.76	11.92

## Data Availability

The data presented in this study are available on request from the corresponding author. The data are not publicly available due to privacy issues.

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
