# Peer review of "Machine Learning Model-Based Simple Clinical Information to Predict Decreased Left Atrial Appendage Flow Velocity"

_jpm, 2022, doi:10.3390/jpm12030437_

Round 1
Reviewer 1 Report
The paper entitled "Machine Learning Model Based Simple Clinical Information to Predict Decreased Left Atrial Appendage Flow Velocity" is an interesting paper that covers at topic of interest for clinicians that could require a simple applicable method to screen patients with decreased left atrial appendage flow velocity (LAAV). Therefore, they investigated the feasibility and accuracy of machine learning model to predict LAAV. The authors try to show a machine learning model based on simple clinical information that explicitly could predict left atrial appendage flow velocity, showing a practical application for screening of decreased LAAV.
Some elements for discussion: Sample size or statistical power need to be presented…not clear addition on the next version of the paper. We could suspect the sample size is not big enough to have reproductible results, the retrospective nature of the study limits the value of the paper and its mention now on the limitations section
I still think the same about the retrospective paper value
Author Response
Response to Reviewer 1 Comments
Point 1: Sample size or statistical power need to be presented…not clear addition on the next version of the paper. We could suspect the sample size is not big enough to have reproductible results, the retrospective nature of the study limits the value of the paper and its mention now on the limitations section.
I still think the same about the retrospective paper value
Response: Thanks for the comment.
The sample size of the training set has been recognized as one of the most important categories on the design and performance of models. Model-based approaches provided sample size estimates based on characteristics of the algorithm, allowed classification error upon generalization, and acceptable confidence in that error. In our models, the number of the number of features was 12, and the allowed classification error was 0.1. By the method proposed by Haykin1, 120 training samples (12/0.1) were sufficient, which was consistent with recommendation of Sengupta2. The sample size in our training set was more than 120, and as mentioned in discussion (page 9, line 21-24), the sample-to-feature was about 70 in present study. So the sample size is relatively sufficient. There are previous studies with similar sample sizes. For example, Ma´rton Tokodi et al.3 had developed machine learning models to predict mortality of patients undergoing cardiac resynchronization therapy. In their study, the number of features was thirty-three and the sample size was 1510 (the sample-to-feature was about 50). In another similar study4, the number of features was 71 and the sample size was 424 (the sample-to-feature was 6).
According to the recommendations of previous studies2, the parameters to validate the performance of machine learning models are AUC, accuracy, F1 score, sensitivity, and specificity, which were reported in the results (table 3 and table 4).
We admit that the retrospective nature of the study limits the value of the paper, which has been mentioned in the revised manuscript. However, supervised machine learning methods need annotated data to generate efficient models, and it was difficult to obtaining annotated data from prospective studies. Therefore, most studies related to machine learning models are retrospective studies, and previous studies5-7 had proved that on retrospective data, individual machine learning models could play a role in the diagnosis of diseases. In the future we will establish a prospective cohort to validate the results of this model.
Reference
- Balki I, Amirabadi A, Levman J, et al. Sample-Size Determination Methodologies for Machine Learning in Medical Imaging Research: A Systematic Review. Can Assoc Radiol J. 2019;70(4):344-353.
- Sengupta PP, Shrestha S, Berthon B, et al. Proposed Requirements for Cardiovascular Imaging-Related Machine Learning Evaluation (PRIME): A Checklist: Reviewed by the American College of Cardiology Healthcare Innovation Council. JACC Cardiovasc Imaging. 2020;13(9):2017-2035.
- Tokodi M, Schwertner WR, Kovacs A, et al. Machine learning-based mortality prediction of patients undergoing cardiac resynchronization therapy: the SEMMELWEIS-CRT score. Eur Heart J. 2020;41(18):1747-1756.
- Zhang J, Zhu H, Chen Y, et al. Ensemble machine learning approach for screening of coronary heart disease based on echocardiography and risk factors. BMC Med Inform Decis Mak. 2021;21(1):187.
- Isci S, Kalender DSY, Bayraktar F, Yaman A. Machine Learning Models for Classification of Cushing's Syndrome Using Retrospective Data. IEEE J Biomed Health Inform. 2021;25(8):3153-3162.
- Myung Y, Jeon S, Heo C, et al. Validating machine learning approaches for prediction of donor related complication in microsurgical breast reconstruction: a retrospective cohort study. Sci Rep. 2021;11(1):5615.
- Gardezi SJS, Elazab A, Lei B, Wang T. Breast Cancer Detection and Diagnosis Using Mammographic Data: Systematic Review. J Med Internet Res. 2019;21(7):e14464.
Reviewer 2 Report
The authors have addressed all my concerns, thank you.
I have one additional minor comment: in Tables 1 and 2 subsections named "Anticoagulant" and "No anticoagulant" may better be described as "Antithrombotic therapy" and "No antithrombotic therapy".
Congratulations on an interesting paper!
Author Response
Response to Reviewer 2 Comments
Point 1: I have one additional minor comment: in Tables 1 and 2 subsections named "Anticoagulant" and "No anticoagulant" may better be described as "Antithrombotic therapy" and "No antithrombotic therapy".
Response: Thanks for the comment. As suggested by the reviewer, we have revised Table 1 and Table 2 as follows:
“Table 1. Baseline characteristics of the records of the enrolled patients with decreased LAAV and normal LAAV.
|
ALL N=1039 |
Normal LAAV N=914 |
Decreased LAAV N=125 |
P value |
Age(y) |
62(54-69) |
62(54-68) |
64(54-71.5) |
0.022 |
Gender(male) |
757(72.9%) |
663(72.5%) |
94(75.2%) |
0.530 |
BMI(Kg/m2) |
26(23-28) |
25.8(23.8-28.0) |
26.9(24.4-29.3) |
0.010 |
Persistent AF |
336(32.3%) |
253(27.7%) |
83(66.4%) |
<0.001 |
Antithrombotic therapy |
|
|
|
<0.001 |
No antithrombotic therapy |
528(50.8%) |
478(52.3%) |
50(40.0%) |
|
Aspirin |
130(12.5%) |
113(12.4%) |
17(13.6%) |
|
Clopidogrel |
11(1.1%) |
8(0.9%) |
3(2.4%) |
|
Dual antiplatelet |
21(2.0%) |
18(2.0%) |
3(2.4%) |
|
Warfarin |
95(9.1%) |
76(8.3%) |
19(15.2%) |
|
NOAC |
254(24.5%) |
221(24.2%) |
33(26.4%) |
|
Heart failure |
108(10.4%) |
67(7.3%) |
41(32.8%) |
<0.001 |
Hypertension |
553(53.2%) |
474(51.9%) |
79(63.2%) |
0.017 |
Diabetes |
239(23.0%) |
200(21.9%) |
39(31.2%) |
0.020 |
Stroke |
154(14.8%) |
127(13.9%) |
27(21.6%) |
0.023 |
Vascular disease |
92(8.9%) |
83(9.1%) |
9(7.2%) |
0.487 |
NT-pro BNP (pg/ml) |
319(116-770) |
266(103-635) |
997(590.5-1862) |
<0.001 |
LA diameter (mm) |
39(35-43) |
38(35-42) |
43(40-47.5) |
<0.001 |
*NOAC: New oral anticoagulants, include dabigatran and rivaroxaban; LA diameter: left atrial diameter.
Table 2. The characteristics of training set and testing set.
Feature |
Training set N=834 |
Testing set N=205 |
P value |
Age(y) |
62(54-69) |
61(54-68.5) |
0.357 |
Gender(male) |
592(71.0%) |
165(80.5%) |
0.006 |
BMI(Kg/m2) |
26(23.9-28.1) |
26(24.2-28.2) |
0.772 |
Persistent AF |
264(31.7%) |
72(35.1%) |
0.342 |
Antithrombotic therapy |
|
|
0.68 |
No antithrombotic therapy |
426(51.2%) |
102(49.8%) |
|
Aspirin |
104(12.5%) |
26(12.7%) |
|
Clopidogrel |
10(1.2%) |
1(0.5%) |
|
Dual antiplatelet |
19(2.3%) |
2(1%) |
|
Warfarin |
72(8.6%) |
23(11.2%) |
|
NOAC |
203(24.3%) |
51(24.8%) |
|
Heart failure |
82(9.8%) |
26(12.7%) |
0.231 |
Hypertension |
449(53.8%) |
104(50.7%) |
0.425 |
Diabetes |
191(22.9%) |
48(23.4%) |
0.876 |
Stroke |
124(14.9%) |
30(14.6%) |
0.933 |
Vascular disease |
78(9.4%) |
14(6.8%) |
0.255 |
NT-pro BNP (pg/ml) |
309(111-725) |
401(150-885) |
0.061 |
LA size(mm) |
39(35-43) |
39(36-42) |
0.446 |
LAAV (cm/s) |
46(33-60.25) |
46(36-42) |
0.727 |
Decreased LAAV |
97(11.6%) |
28(13.7%) |
0.424 |
”
Round 2
Reviewer 1 Report
Congratulations to the authors for addressing the questions ... I feel there's an unmet need for prospective studies, these kinds of studies are needed to get these techniques forward
This manuscript is a resubmission of an earlier submission. The following is a list of the peer review reports and author responses from that submission.
Round 1
Reviewer 1 Report
In their manuscript titled “Machine Learning Model Based Simple Clinical Information to Predict Decreased Left Atrial Appendage Flow Velocity” the authors describe three machine learning models predicting low left atrial appendage flow velocity. Generally, this work is interesting and novel, and provides new clinically important information. However, there are few issues that should be addressed by the authors:
- The authors state “the data was randomly split into a training set (70%) and a test set (30%)”. Please provide the allocation method.
- Which method was used for the elimination of dimensional impact on model performance?
- Although the authors describe that the training set and test set were used, the results are not presented for two data sets separately. The authors should provide clear comparison of the prediction models in two data sets.
- The authors may like to provide calibration plots (observed vs predicted risk) for every model used.
- In the Limitations the authors state “In the future study, we will improve the model by including more Imaging indicators…”. I would recommend deleting this sentence or providing these data in the current manuscript.
- The Conclusion should contain specific findings (more detailed)
- The manuscript requires English language revision.
Reviewer 2 Report
The paper entitled "Machine Learning Model Based Simple Clinical Information to Predict Decreased Left Atrial Appendage Flow Velocity" is an interesting paper that covers at topic of interest for clinicians that could require a simple applicable method to screen patients with decreased left atrial appendage flow velocity (LAAV). Therefore, they investigated the feasibility and accuracy of machine learning model to predict LAAV. The authors try to show a machine learning model based on simple clinical information that explicitly could predict left atrial appendage flow velocity, showing a practical application for screening of decreased LAAV.
Some elements for discussion: Sample size or statistical power need to be presented. We could suspect the sample size is not big enough to have reproducible results, the retrospective nature of the study limits the value of the paper
Reviewer 3 Report
The authors evaluate the performance of 3 machine learning classifiers (Support Vector Machine, Random Forest and k-Nearest Neighbor) to predict Left Atrial Appendage Flow Velocity (LAAV) in a dataset with 1039 patients, where 125 were diagnosed with this disorder. In test data, RF model perform best (R=0.608, AUROC =0.89) among three models.
My comments are:
1.-Review and correct errors (omissions, typos, grammar, among others) in the writing.
2.-Add references and give more details of the following statement: “Covariates for the machine leaning model were chosen based on prior literature review and clinical judgment, focusing on variables which were expected to affect the LAAV.”
3.-In section 2.5, you mention that you carried out regression models, however, from the reported metrics it is inferred that you also carried out classification. This point must be clarified, and where appropriate, explain in detail how this process was carried out. In general, the experimental procedure and the presentation of the results require improvement.
4.-In the same section, add references to the following statement: “…because they are used widely and successfully in regression model.”. Review and add the necessary references throughout the document.
5.-"Information gain values ​​were used to interpret the model.” What model? What is the information gain? What is the use of interpreting a model with the information gain? You do not include this interpretation and do not mention this metric again in the rest of the document.
6.-Improve the wording of section 3. Some statements are incomplete, e.g. “Continuous were presented as median…”. You do not specify why you did what you mention there, nor do you give more details. Which data are continuous and which are categorical? In general, you assume that the reader will understand the document as it is written.
7.-The number of excluded cases does not agree with the diagram in Figure 1 (110 reported).
8.-834 is not exactly 80% of 1039.
9.-Your dataset is clearly unbalanced, how do you justify the use of the metrics reported in your study?
10.-Include the acceptable values or found so far of the metrics used in your study and compare them with your results.
